# In Vitro Rumen Fermentation of Coconut, Sugar Palm, and Durian Peel Silages, Prepared with Selected Additives

Waroon Khota [1], Paiwan Panyakaew [1], Piyawit Kesorn [1], Pongsatorn Gunun [1], Rattikan Suwannasing [1], Thachawech Kimprasit [1], Premsak Puangploy [2], Ketinun Kittipongpittaya [2], Anusorn Cherdthong [3], Suwit Thip-uten [4], Pakpoom Sawnongbua [4,*] and Chatchai Kaewpila [1,*]

1 Faculty of Natural Resources, Rajamangala University of Technology Isan, Sakon Nakhon 47160, Thailand; waroon.kh@rmuti.ac.th (W.K.); paiwan.pa@rmuti.ac.th (P.P.); piyawit.ke@rmuti.ac.th (P.K.); pongsatorn.gu@rmuti.ac.th (P.G.); rattikan.su@rmuti.ac.th (R.S.); thachawech.ki@rmuti.ac.th (T.K.)
2 Faculty of Agro-Industry, King Mongkut's University of Technology North Bangkok, Prachinburi 25230, Thailand; premsak.p@agro.kmutnb.ac.th (P.P.); ketinun.k@agro.kmutnb.ac.th (K.K.)
3 Faculty of Agriculture, Khon Kaen University, Khon Kaen 40002, Thailand; anusornc@kku.ac.th
4 Faculty of Agricultural Technology, Sakon Nakhon Rajabhat University, Sakon Nakhon 47000, Thailand; suwit@snru.ac.th
* Correspondence: pakpoom@snru.ac.th (P.S.); chatchai.ka@rmuti.ac.th (C.K.); Tel.: +66-84-262-9518 (C.K.)

**Abstract:** Understanding the nutritive values of fruit peel residues could expand our feed atlas in sustaining livestock production systems. This study aimed to investigate the effects of lactic acid bacteria (LAB), cellulase enzyme, molasses, and their combinations on the fermentation quality and in vitro digestibility of coconut peel (CCP), sugar palm peel (SPP), and durian peel (DRP) silage. The CCP, SPP, and DRP were ensiled in a small-scale silo without additive (control), and with LAB strain TH14 (TH14), molasses, or *Acremonium* cellulase (AC) using a small-scale silage preparation technique according to a completely randomized design. All fresh peels had sufficient factors for ensiling such as moisture content (78–83%), water-soluble carbohydrates (WSC, 4.20–4.61% dry matter (DM)), and epiphytic LAB population ($10^4$–$10^5$ colony-forming units (cfu)/g fresh matter (FM)). However, aerobic bacteria counts were high ($10^7$–$10^9$ cfu/g FM). The fiber content of these fruit peels was high, with lignin abundances ranging from 9.1–21.8% DM and crude protein was low (2.7–5.4% DM). After ensiling, the pH values of the silage were optimal ($\leq 4.25$) and lower ($p < 0.01$) for SPP silage. The addition of molasses+TH14, molasses+AC, and molasses+TH14+AC has the potential to enhance fermentation characteristics and improve chemical composition. Silages treated with molasses alone improved the in vitro digestibility of tropical fruit peels. The residue of tropical fruits has the potential to be used as an alternative feed source for ruminants. Adding molasses, TH14, and AC during silage preparation could improve its nutritive value and digestibility.

**Keywords:** tropical fruit peels silage; digestibility; silage additives

## 1. Introduction

Coconut (*Cocos nucifera* L.), Sugar palm (*Borassus flabillifer* L.), and Durian (*Durio zibethinus* Murr.) are popular tropical fruits grown widely in tropical countries including Thailand. For young coconut and sugar palm and ripe durian, only 20% to 30% of the fruit is suitable for human consumption. The remaining 70% to 80% comprises the fruit peel, which is typically discarded as waste [1–3]. The possible ways of agricultural by-product utilization include composting, which is beneficial to the environment, provides nutrient-rich soil amendments, and can generate income for farmers. On the other hand, in animal husbandry, these peels have the potential to serve as valuable feed sources for animals. They can reduce agricultural waste and provide farmers with an alternative feed option that can lower production costs. However, these fruit residues have high moisture content and are extremely susceptible to spoilage.

Silage is the most suitable technology for preserving high-moisture material [4], and it has become a widely used method for preserving fresh forage as ruminant feed worldwide [5]. The aim of making silage is to produce a stable feed that recovers a high amount of dry matter (DM), energy, and highly digestible nutrients compared to the fresh crop [6]. Silage additives such as bacteria inoculants, fibrolytic enzymes, and molasses have significantly contributed to enhancing the quality of silage and improving nutrient digestibility [7]. Previous studies have reported that the addition of lactic acid bacteria (LAB) improved the ensiling process by decreasing the pH value and promoting lactic acid production [8–10]. The addition of fibrolytic enzymes in ensiling materials has the potential to enhance the breakdown of fiber, leading to an increase in water-soluble carbohydrates (WSC) contents and the production of lactic acid [11–13]. Kaewpila et al. [14] and Gül [15] reported that the addition of molasses into ensiling materials has a positive effect on silage fermentation quality. The application of LAB inoculant combined with cellulase, or molasses enhances the ensiling process and in vitro digestibility [16,17].

However, there is very limited information available on silage fermentation and in vitro digestibility of tropical fruit peel, especially coconut, sugar palm, and durian, when treated with silage additives in the tropics. The objectives of this study were to determine the effects of LAB, cellulase enzyme, molasses, and their combinations on the fermentation quality and in vitro digestibility of tropical fruit peels silage.

## 2. Materials and Methods

### 2.1. Ensiling Process and Experimental Design

In May 2022, we obtained 10 kg each of the young coconut peel (CCP) at the harvested stage of 180–200 days after flowering, the young sugar palm peel (SPP) at 75–80 days after flowering, and the ripe durian peel (DRP) at 110–120 days after flowering from the local market in Khon Kaen province, Thailand. After being collected, they were immediately chopped into 1 to 2 cm pieces using a forage chopper (Supachai, Kanchanaburi, Thailand).

The experimental design was a $3 \times 8$ factorial arrangement in a completely randomized design (fruit peel $\times$ additives) with 4 silo replications. A locally selected strain *Lactobacillus casei* TH14 (TH14) [18], a commercial *Acremonium* cellulase enzyme (AC; Meiji Seika Pharma Co., Ltd., Tokyo, Japan), sugar cane molasses (molasses), and their mixture were used as silage additives. The TH14 inoculum was inoculated at the rate of $1.0 \times 10^5$ cfu/g of fresh matter (FM). The AC and molasses were added at 0.01% and 2% of FM, respectively. Three ensiling materials (CCP, SPP, and DRP) were treated with eight additives: control (untreated), TH14, AC, molasses, TH14+AC, molasses+TH14, molasses+AC, and TH14+AC+molasses.

Eight hundred grams of each chopped fruit peel were mixed well with the assigned additives, and two hundred grams were packed into each bag silo with laminated nylon and polyethylene (Hiryu KN, Asahikasei, Tokyo, Japan), and sealed using a vacuum sealer (SQ–303, Asahi Kasei Pax Corp., Tokyo, Japan). All silos were kept at ambient temperature (25 to 37 °C). After 30 days of ensiling, all bag silos were opened for evaluation of fermentation end-products, chemical composition, microorganism population, and in vitro ruminal digestibility.

### 2.2. Fermentation Analysis

The silage quality and organic acid products were determined using the cold extract as described by Cai [19]. Ten grams of wet silage sample were blended with 90 mL of distilled water and incubated at 4 °C in a refrigerator for 12 h. After incubation, the extracted sample was warmed to 25 °C, and then the pH value was measured using a pH meter (pH meter FiveGo, Mettler-Toledo GmbH, Greifensee, Switzerland). The lactic acid, acetic acid, propionic acid, butyric acid, and total alcohol (methanol, ethanol, iso-propanol, and 1-butanol) concentrations were analyzed using gas chromatography (Nexis GC-2030, Shimadzu Co., Kyoto, Japan) equipped with a capillary column (DB-WAX 30 m, 0.25 mm, 0.25 μm, Agilent Technologies, Inc., Santa Clara, CA, USA) and flame ionization detector.

The ammonia-N concentration was determined using a UV-visible spectrophotometer (UV/VIS Spectrometer, PG Instruments Ltd., London, UK) according to the methods of Fawcett and Scott [20].

### 2.3. Chemical Analysis

Adequate samples of fresh materials and their silages were dried in a hot air oven at 60 °C for 48 h and then ground to pass through a 1 mm mesh screen using a sample mill (MF 10 basic, IKA, Staufen, Germany). The DM content was analyzed using the dry method in a hot air oven at 100 °C for 24-h. The organic matter (OM) and ether extract (EE) contents were analyzed following the standard methods of AOAC [21] (942.05 and 920.39, respectively). The CP content was analyzed using the combustion technique with an N-analyzer (828 Series, LECO, St. Joseph, MI, USA) standardized with EDTA and using 6.25 as the factor for CP conversion. The neutral detergent fiber (NDF) and acid detergent fiber (ADF) contents were analyzed using a fiber analyzer (ANKOM 200, ANKOM Technology, Macedon, NY, USA). For NDF analysis, alpha-amylase (2500 U/mg, Sigma-Aldrich, St Louis, MO, USA) and sodium sulfite (crystalline, 98.0% assay, Kemaus, NSW, Australia) were used to digest starch and protein, respectively. The acid detergent lignin (ADL) was analyzed by solubilization with a 72% sulfuric acid solution [22]. The WSC contents were measured by HPLC methods as described by Cai [16].

### 2.4. Microorganism Analysis

The microorganism populations in ensiling materials and their silages were counted using the total plate-count technique [8,23]. Briefly, 10 g of FM sample was blended with 90 mL of sterilized 0.85% NaCl and serially diluted at $10^{-1}$ to $10^{-5}$. Then, 20 μL of each dilution was spread on prepared agar plates. The colonies of LAB were counted on MRS agar (Difco) after incubation at 30 °C for 48 h in an aerobic box (Sugiyamagen Ltd., Tokyo, Japan). Coliform bacteria and aerobic bacteria were counted on blue-light agar (Nissui-seiyaku Ltd., Tokyo, Japan) and nutrient agar (Difco), respectively, after incubation at 30 °C for 72 h under aerobic conditions. Yeasts and molds were observed on potato dextrose agar (Nissui-seiyaku Ltd., Tokyo, Japan) and distinguished by observing the cell morphology after 3 to 7 days of incubation.

### 2.5. In Vitro Ruminal Digestibility and Fermentation Products

The silage samples were investigated for in vitro dry matter (IVDMD) and organic matter digestibility (IVOMD) after 24 and 48 h of incubation using a gas production technique. The artificial saliva was prepared according to Makkar et al. [24]. Two lactating Holstein cows in the peak intake of lactation period were used as ruminal fluid donors. The cattle were fed with a fermented total mixed ration containing rice straw and concentrate feed at a ratio of 50:50 on a DM basis and had free access to fresh drinking water.

The rumen fluid was collected before the morning feeding by a stomach-tube sucker. The first 500 mL of collected rumen fluid was discarded to avoid saliva contamination, according to Muizelaar et al. [25]. After correction, the rumen fluid was filtered through four layers of cheesecloth and mixed with artificial saliva at a ratio of 1:4 under a $CO_2$-flushed atmosphere. Then, 0.5 g of ground silage sample was weighed into a 50 mL serum bottle with three duplications for each replicate of the eight treatments, and three bottles without sample were used as blanks. The bottles were closed with rubber stoppers and aluminum caps and 40 mL of rumen medium was injected into each bottle using a 60 mL syringe (Nipro (Thailand) Co., Ltd., Phra-Nakhon-Si-Ayutthaya, Thailand) with an 18 gauge × 1.5-inch needle (Nipro (Thailand) Co., Ltd.).

All bottles were incubated at 39 °C with shaking at 120 rpm (Innova 40, Hamburg, Germany). The gas produced was released from the bottle every 2 h interval. After 24 and 48 h of incubation, the residual samples were filtered through a glass filter crucible (ROBU, GmbH, Hattert, Germany), dried at 100 °C in a hot air oven for 24 h, and weighed for in IVDMD determination. The dried residues were ashed at 550 °C for 3 h for IVOMD

calculation. The pH, volatile fatty acids (VFAs), and ammonia-N were determined using the same method described above for silages.

### 2.6. Statistical Analysis

The data obtained from silages were analyzed using an ANOVA procedure of SAS Version 6.12 (SAS Institute Inc., Cary, NC, USA). The statistical model is as follows:

$$Y_{ijk} = \mu + \alpha_i + \beta_j + \alpha\beta_{ij} + \varepsilon_{ijk,} \tag{1}$$

where $Y_{ijk}$ = observation, $\mu$ = overall mean, $\alpha_i$ = tropical fruit peels effect (i = 3), $\beta_j$ = additive effect (j = 8), $\alpha\beta_{ij}$ = tropical fruit peels × additive, and $\varepsilon_{ijk}$ = error. The difference among treatment means were compared by Duncan's test at $p \leq 0.05$ [26].

## 3. Results

### 3.1. Tropical Fruit Peel Materials

Prior to ensiling, the fresh CCP, SPP, and DRP samples had similar microorganism populations, with LAB ranging from $10^4$ to $10^5$ cfu/g FM, coliform bacteria and aerobic bacteria ranging from $10^7$ to $10^9$ cfu/g FM, and yeast and molds ranging from $10^6$ to $10^9$ cfu/g FM (Table 1). The DM content of fresh tropical fruit peels ranged from 17.4 to 22.3%, CP from 2.7 to 5.4% on DM, NDF from 55.4 to 68.5% on DM, ADF from 43.5 to 55.2% on DM, ADL from 9.1 to 21.8% on DM, and WSC from 4.2 to 4.6% on DM.

**Table 1.** Microbial counts and chemical composition of pre-ensiled materials used in this study.

| Item | CCP | SPP | DRP |
|---|---|---|---|
| LAB, cfu/g FM | $2.2 \times 10^5$ | $1.5 \times 10^5$ | $7.3 \times 10^4$ |
| Coliform bacteria, cfu/g FM | $3.9 \times 10^7$ | $1.4 \times 10^9$ | $4.2 \times 10^7$ |
| Aerobic bacteria, cfu/g FM | $9.3 \times 10^8$ | $1.4 \times 10^9$ | $5.5 \times 10^7$ |
| Yeasts, cfu/g FM | $1.4 \times 10^9$ | $1.4 \times 10^8$ | $1.4 \times 10^8$ |
| Molds, cfu/g FM | $4.6 \times 10^7$ | $1.5 \times 10^9$ | $3.0 \times 10^6$ |
| DM, % | 19.25 | 17.37 | 22.29 |
| OM, % on DM | 95.12 | 95.35 | 94.00 |
| CP, % on DM | 2.74 | 5.40 | 5.02 |
| EE, % on DM | 0.60 | 0.97 | 0.77 |
| NDF, % on DM | 65.94 | 55.35 | 68.49 |
| ADF, % on DM | 55.23 | 43.47 | 47.13 |
| ADL, % on DM | 21.83 | 9.13 | 9.34 |
| WSC, % on DM | 4.20 | 4.23 | 4.61 |

cfu, colony forming unit; FM, fresh matter; LAB, lactic acid bacteria; DM, dry matter; OM, organic matter; CP, crude protein; EE, ether extract; NDF, neutral detergent fiber; ADF, acid detergent fiber; ADL, acid detergent lignin; WSC, water soluble carbohydrate; CCP, coconut peel; SPP, sugar palm peel; DRP, durian peel.

### 3.2. Fermentation Quality of Tropical Fruit Peel Silages

The A, B, and A × B significantly ($p < 0.01$) influenced silage pH and content of organic acids, except butyric acid ($p > 0.05$) content (Table 2). The A and A × B were also significantly ($p < 0.05$) influenced DM content, and A significantly ($p < 0.05$) influenced ($p < 0.05$) ammonia-N content. All silages were well preserved with a relatively low pH (<4.2), butyric acid (<0.7 g/kg DM), and ammonia-N (0.1 g/kg DM), as well as high lactic acid content (>53.9 g/kg DM). The SPP silages had significantly ($p < 0.01$) lower pH and higher ($p < 0.01$) acetic acid content than CCP and DRP silages. The DM content, lactic acid, and ammonia-N of DRP silages were higher ($p < 0.01$) than CCP and SPP silages, whereas total alcohol concentration was lower ($p < 0.01$). Silages treated with molasses+TH14+AC had lower ($p < 0.01$) pH and propionic acid, and higher ($p < 0.05$) lactic acid content than other treatments. The highest ($p < 0.01$) acetic acid contents were found in AC and molasses+AC treatments. Compared with other treatments, AC and TH14+AC

additions increased ($p < 0.01$) the total alcohol contents in silages. The DM, butyric acid, and ammonia-N were not different ($p > 0.05$) in all treatments.

**Table 2.** Ensiling characteristics of tropical fruit peel silage after 30 days of fermentation.

| Item | DM | pH | Lactic Acid | Acetic Acid | Propionic Acid | Butyric Acid | Ammonia-N | Total Alcohol |
|---|---|---|---|---|---|---|---|---|
| | % | | | | g/kg DM | | | |
| Fruit peels means | | | | | | | | |
| CCP | 17.37 [c] | 4.19 [a] | 50.54 [c] | 9.60 [c] | 0.07 [a] | 0.458 | 0.087 [b] | 61.30 [a] |
| SPP | 18.36 [b] | 3.46 [c] | 76.16 [b] | 14.84 [a] | 0.01 [b] | 0.615 | 0.091 [ab] | 51.42 [b] |
| DRP | 20.72 [a] | 3.77 [b] | 97.36 [a] | 11.83 [b] | 0.01 [b] | 0.365 | 0.097 [a] | 30.20 [c] |
| Additive means | | | | | | | | |
| Control | 18.87 | 3.89 [b] | 47.16 [d] | 9.44 [c] | 0.103 [a] | 0.741 | 0.102 | 44.71 [bcd] |
| TH14 | 18.41 | 3.83 [c] | 69.18 [b] | 8.59 [c] | 0.060 [ab] | 0.311 | 0.088 | 40.51 [cd] |
| AC | 18.41 | 3.80 [dc] | 86.03 [b] | 16.30 [a] | 0.011 [bc] | 0.210 | 0.092 | 54.98 [a] |
| TH14+AC | 18.46 | 3.72 [e] | 85.71 [b] | 13.75 [b] | 0.009 [bc] | 0.326 | 0.091 | 56.03 [a] |
| Molasses | 19.47 | 3.96 [a] | 49.95 [c] | 10.86 [c] | 0.013 [bc] | 0.187 | 0.093 | 38.15 [d] |
| Molasses+TH14 | 19.00 | 3.79 [cd] | 77.63 [b] | 10.75 [c] | 0.002 [c] | 0.266 | 0.090 | 45.77 [bcd] |
| Molasses+AC | 19.04 | 3.74 [de] | 75.53 [b] | 16.20 [a] | 0.007 [bc] | 0.212 | 0.089 | 49.09 [abc] |
| Molasses+TH14+AC | 18.87 | 3.72 [e] | 106.28 [a] | 10.82 [c] | 0.006 [c] | 1.581 | 0.088 | 51.86 [ab] |
| SEM | 0.497 | 0.035 | 9.521 | 1.297 | 0.029 | 0.790 | 0.007 | 4.929 |
| Significance of main effect and interaction | | | | | | | | |
| Fruit peels (A) | <0.001 | <0.001 | <0.001 | <0.001 | <0.001 | 0.815 | 0.026 | <0.001 |
| Additives (B) | 0.135 | <0.001 | <0.001 | <0.001 | 0.001 | 0.376 | 0.322 | 0.001 |
| A × B | 0.049 | 0.001 | <0.001 | <0.001 | <0.001 | 0.446 | 0.648 | 0.002 |

[a–e] Means within columns with difference superscript letters differ at $p < 0.05$; Values are means of three silage samples; DM, dry matter; ND, not detected; TH14, *Lactobacillus casei*; AC, *Acremonium* cellulase; CCP, coconut peel; SPP, sugar palm peel; DRP, durian peel; SEM, standard error of the mean.

### 3.3. Chemical Composition of Tropical Fruit Peel Silages

The fruit peel means (A), additives (B), and A × B significantly ($p < 0.05$) influenced all chemical compositions of silages (Table 3). The CP content of SPP silages was significantly ($p < 0.01$) higher, whereas OM and EE were lower ($p < 0.01$) than the other two silages. The lowest ($p < 0.01$) NDF, ADF, and ADL were found in DRP silages. The OM content of the TH14-treatment was significantly ($p < 0.01$) higher than the control and other treatments. When silages were treated with molasses+AC and molasses+TH14+AC, the CP content was significantly ($p < 0.01$) higher, and the NDF and ADF content were significantly ($p < 0.01$) lower than other treatments. The lowest ($p < 0.01$) ADL was found in molasses+TH14 treatment.

**Table 3.** Chemical composition of tropical fruit peel silage after 30 days of fermentation.

| Item | OM | CP | EE | NDF | ADF | ADL |
|---|---|---|---|---|---|---|
| | % on DM | | | | | |
| Fruit peels means | | | | | | |
| CCP | 94.14 [b] | 3.13 [c] | 0.83 [a] | 76.91 [a] | 66.94 [a] | 27.81 [a] |
| SPP | 92.26 [c] | 6.54 [a] | 0.59 [b] | 64.94 [b] | 47.18 [b] | 8.91 [b] |
| DRP | 94.54 [a] | 5.72 [b] | 0.81 [a] | 63.96 [b] | 44.20 [c] | 8.92 [b] |
| Additive means | | | | | | |
| Control | 93.94 [ab] | 4.55 [c] | 0.77 [bc] | 71.69 [a] | 55.46 [a] | 15.87 [abc] |
| TH14 | 94.01 [a] | 4.71 [c] | 0.58 [bcd] | 71.70 [a] | 55.31 [a] | 15.16 [bcd] |
| AC | 93.75 [bcd] | 4.81 [c] | 0.83 [b] | 69.98 [ab] | 53.82 [a] | 16.28 [a] |
| TH14+AC | 93.79 [abc] | 5.39 [ab] | 1.18 [a] | 69.94 [ab] | 54.71 [a] | 16.12 [ab] |
| Molasses | 93.60 [cd] | 5.15 [b] | 0.81 [b] | 69.31 [b] | 51.89 [b] | 14.98 [cd] |
| Molasses+TH14 | 93.53 [de] | 5.30 [ab] | 0.51 [d] | 68.45 [b] | 51.22 [b] | 13.71 [e] |
| Molasses+AC | 93.32 [ef] | 5.54 [a] | 0.72 [bcd] | 64.49 [c] | 50.31 [bc] | 14.98 [cd] |
| Molasses+TH14+AC | 93.23 [f] | 5.59 [a] | 0.52 [cd] | 63.28 [c] | 49.46 [c] | 14.61 [de] |
| SEM | 0.137 | 0.204 | 0.142 | 1.129 | 0.983 | 0.595 |
| Significance of main effect and interaction | | | | | | |
| Fruit peels (A) | <0.001 | <0.001 | 0.002 | <0.001 | <0.001 | <0.001 |

**Table 3.** *Cont.*

| Item | OM | CP | EE | NDF | ADF | ADL |
|---|---|---|---|---|---|---|
| | | | **% on DM** | | | |
| Additives (B) | <0.001 | <0.001 | <0.001 | <0.001 | <0.001 | <0.001 |
| A × B | 0.048 | 0.015 | <0.001 | <0.001 | <0.001 | 0.004 |

[a–f] Means within columns with difference superscript letters differ at *p* < 0.05; Values are means of three silage samples; OM, organic matter; CP, crude protein; EE, ether extract; NDF, neutral detergent fiber; ADF, acid detergent fiber; ADL, acid detergent lignin; TH14, *Lactobacillus casei*; AC, *Acremonium* cellulase; CCP, coconut peel; SPP, sugar palm peel; DRP, durian peel; SEM, standard error of the means.

### 3.4. Microbial Populations of Tropical Fruit Peel Silages

The A, B, and A × B significantly (*p* < 0.05) influenced aerobic the bacteria count, and the A was also significantly different (*p* < 0.05) in the LAB count of silages (Table 4). Coliform bacteria, yeast, and mold were not detected in all silages. The LAB populations of CCP and DRP silages were dominated (7.17 log10 cfu/g FM), which was greater than that of SPP silages (*p* < 0.05). The aerobic bacteria population in SPP silages was lower than that of CCP and DRP silages. The silages treated with AC had the lowest (*p* < 0.05) aerobic bacteria population compared to the control and other treatments.

**Table 4.** Microbial counts tropical fruit peel silage after 30 days of fermentation.

| Item | Microorganism (log10 cfu/g FM) | |
|---|---|---|
| | **Lactic Acid Bacteria** | **Aerobic Bacteria** |
| Fruit peels means | | |
| CCP | 7.17 [a] | 6.49 [a] |
| SPP | 6.55 [b] | 4.05 [c] |
| DRP | 6.78 [ab] | 4.88 [b] |
| Additive means | | |
| Control | 7.18 | 6.02 [a] |
| TH14 | 6.84 | 5.60 [b] |
| AC | 7.16 | 3.49 [e] |
| TH14+AC | 6.99 | 4.86 [d] |
| Molasses | 6.01 | 5.50 [bc] |
| Molasses+TH14 | 6.89 | 5.45 [bc] |
| Molasses+AC | 6.72 | 4.94 [d] |
| Molasses+TH14+AC | 6.88 | 5.26 [c] |
| SEM | 0.469 | 0.178 |
| Significance of main effect and interaction | | |
| Fruit peels (A) | 0.033 | <0.001 |
| Additives (B) | 0.095 | <0.001 |
| A × B | 0.268 | <0.001 |

[a to c] Means within columns with difference superscript letters differ at *p* < 0.05; Values are means of three silage samples; cfu, colony forming unit; FM, fresh matter; ND, not detected; TH14, *Lactobacillus casei* TH14; AC, *Acremonium* cellulase; CCP, coconut peel; SPP, sugar palm peel; DRP, durian peel; SEM, standard error of the means.

### 3.5. In Vitro Digestibility and Fermentation Product of Tropical Fruit Peel Silages

The IVDMD, IVOMD, and rumen fermentation products of tropical fruit peels silage after 30 days of ensiling are shown in Tables 5 and 6 for 24 h and 48 h incubation, respectively. At 24 h after incubation, the A, B, and A × B were significantly (*p* < 0.05) influenced IVDMD, IVOMD, and total VFAs contents. The A was significantly (*p* < 0.01) influenced pH value and ammonia-N content. The SPP and DRP silages had significantly (*p* < 0.01) higher IVDMD, IVOMD, and total VFAs contents, and significantly (*p* < 0.01) lower ammonia-N content than the CCP silages. The pH value was the highest in CCP silages. Silage prepared with molasses had greater (*p* < 0.01) digestibility and total VFAs contents than

the other treatments. The pH and ammonia-N were not significantly ($p > 0.05$) different in all treatments.

**Table 5.** In vitro digestibility and in vitro rumen fermentation characteristics at 24 h of tropical fruit peel silage after 30 days of fermentation.

| Item | IVDMD | IVOMD | pH | Total VFAs | Ammonia-N |
|---|---|---|---|---|---|
| | (%) | (%) | | (mmol/L) | (mg/L) |
| Fruit peels means | | | | | |
| CCP | 12.52 [b] | 12.78 [b] | 7.27 [a] | 35.01 [b] | 175.35 [a] |
| SPP | 53.75 [a] | 55.76 [a] | 7.12 [b] | 71.95 [a] | 125.54 [b] |
| DRP | 54.32 [a] | 56.48 [a] | 7.09 [b] | 69.52 [a] | 129.57 [b] |
| Additive means | | | | | |
| Control | 42.49 [ab] | 42.87 [bc] | 7.06 | 56.78 [bc] | 150.80 |
| TH14 | 41.64 [b] | 42.11 [c] | 7.20 | 59.06 [bc] | 117.31 |
| AC | 35.59 [d] | 37.19 [d] | 7.20 | 57.57 [bc] | 135.94 |
| TH14+AC | 35.65 [d] | 37.60 [d] | 7.14 | 55.71 [bc] | 174.55 |
| Molasses | 44.74 [a] | 45.63 [a] | 7.15 | 68.42 [a] | 139.58 |
| Molasses+TH14 | 43.19 [ab] | 44.73 [ab] | 7.17 | 62.20 [ab] | 137.13 |
| Molasses+AC | 39.02 [c] | 41.59 [c] | 7.18 | 57.81 [bc] | 164.30 |
| Molasses+TH14+AC | 39.26 [c] | 41.67 [c] | 7.15 | 53.02 [c] | 128.26 |
| SEM | 1.324 | 1.196 | 0.077 | 4.346 | 24.695 |
| Significance of main effect and interaction | | | | | |
| Fruit peels (A) | <0.001 | <0.001 | <0.001 | <0.001 | <0.001 |
| Additives (B) | <0.001 | <0.001 | 0.445 | 0.004 | 0.121 |
| A x B | <0.001 | <0.001 | 0.241 | <0.001 | 0.137 |

[a–d] Means within columns with difference superscript letters differ at $p < 0.05$; Values are means of three silage samples; IVDMD, in vitro dry matter digestibility; IVOMD, in vitro organic matter digestibility; TH14, *Lactobacillus casei*; AC, *Acremonium* cellulase; CCP, coconut peel; SPP, sugar palm peel; DRP, durian peel; SEM, standard error of the means.

**Table 6.** In vitro digestibility and in vitro rumen fermentation characteristics at 48 h of tropical fruit peel silage after 30 days of fermentation.

| Item | IVDMD | IVOMD | pH | Total VFAs | Ammonia-N |
|---|---|---|---|---|---|
| | (%) | (%) | | (mmol/L) | (mg/L) |
| Fruit peels means | | | | | |
| CCP | 14.99 [b] | 16.90 [b] | 7.26 [a] | 37.72 [b] | 150.55 [a] |
| SPP | 62.53 [a] | 66.40 [a] | 7.07 [b] | 83.29 [a] | 88.16 [b] |
| DRP | 63.19 [a] | 67.45 [a] | 7.16 [ab] | 83.16 [a] | 82.43 [b] |
| Additive means | | | | | |
| Control | 48.24 [abc] | 49.36 [bcd] | 7.14 | 69.18 [ab] | 93.38 |
| TH14 | 48.30 [abc] | 50.75 [abc] | 7.16 | 68.91 [ab] | 105.22 |
| AC | 43.85 [d] | 47.32 [d] | 7.16 | 74.75 [a] | 107.48 |
| TH14+AC | 43.77 [d] | 47.85 [cd] | 7.13 | 71.08 [ab] | 132.76 |
| Molasses | 50.73 [a] | 53.61 [a] | 7.15 | 73.55 [a] | 78.44 |
| Molasses+TH14 | 49.07 [ab] | 52.36 [ab] | 7.17 | 65.60 [ab] | 104.29 |
| Molasses+AC | 45.27 [cd] | 49.82 [bcd] | 7.11 | 61.16 [b] | 117.10 |
| Molasses+TH14+AC | 46.01 [bcd] | 50.96 [abc] | 7.31 | 60.23 [b] | 117.72 |
| SEM | 1.772 | 1.822 | 0.096 | 6.500 | 28.822 |
| Significance of main effect and interaction | | | | | |
| Fruit peels (A) | <0.001 | <0.001 | 0.002 | <0.001 | <0.001 |
| Additives (B) | <0.001 | 0.001 | 0.299 | 0.069 | 0.459 |
| A × B | 0.017 | 0.096 | 0.193 | 0.019 | 0.693 |

[a–d] Means within columns with difference superscript letters differ at $p < 0.05$; Values are means of three silage samples; IVDMD, in vitro dry matter digestibility; IVOMD, in vitro organic matter digestibility; TH14, *Lactobacillus casei*; AC, *Acremonium* cellulase; CCP, coconut peel; SPP, sugar palm peel; DRP, durian peel; SEM, standard error of the means.

At 48 h after incubation, the A was significantly ($p < 0.01$) influenced IVDMD, IVOMD, pH, total VFAs, and ammonia-N content. The B was significantly ($p < 0.01$) influenced IVDMD and IVOMD. The A × B was significantly ($p < 0.01$) influenced IVDMD and total VFAs, whereas IVOMD, pH, and ammonia-N did not ($p > 0.05$). The SPP and DRP silages had significantly ($p < 0.01$) higher IVDMD, IVOMD, and total VFAs than the CCP silages. Silage treated with molasses had greater ($p < 0.01$) digestibility and acetic acid contents than the other treatments. The pH and ammonia-N contents were not significantly ($p > 0.05$) different in all treatments.

## 4. Discussion

### 4.1. Tropical Fruit Peel Materials

In silage-making practices, microorganisms are significant contributors to the fermentation process especially lactic acid bacteria, as they utilize the sugar content in the ensiling materials to support their growth and produce lactic acid in the silo [27]. The optimal quantity of epiphytic LAB required for producing high-quality silage from tropical grass was determined to be $10^5$ cfu/g FM [28]. In the current study, the population of LAB in all fruit peel materials was ranged from $10^4$ to $10^5$ cfu/g FM (Table 1), which might be adequate for fermenting and producing lactic acid. Whilst the population of coliform bacteria, aerobic bacteria, yeast, and mold were higher ($10^7$ to $10^9$ cfu/g FM) and towered above epiphytic LAB population. These undesirable microbiotas can cause anaerobic spoilage or aerobic spoilage [29]. Thus, to prevent fermentation failures, the addition of silage additives is necessary in the ensiling process. To ensure good quality silage, it is crucial to consider not only the LAB population but also the chemical composition of the ensiling material including the contents of DM and WSC. It is essential that these components are appropriate and present in adequate quantities. [9]. In the present study, the DM content of fruit peels ranged from 17 to 22%, which lower the optimal range (30 to 35%) that reported by Wilkinson [30] and McDonald et al. [27], and the WSC content ranged from 4.20 to 4.61% of DM. The CP content of ensiling materials ranged from 2 to 5% of DM, while the NDF and ADF contents relatively high (55 to 68% and 43 to 55% of DM for NDF and ADF, respectively). The high NDF and ADF contents were not conducive to ensiling fermentation and animal digestion [31].

### 4.2. Fermentation Characteristics of Tropical Fruit Peel after Ensiling

The most common parameters used to evaluate silage fermentation quality are pH, organic acids, alcohol, and ammonia-N contents, as well as the populations of microorganisms. The results show that all silages were well-preserved, with low pH values (3.5 to 4.2) and high lactic acid concentration (Table 2). The DRP silages were higher ($p < 0.01$) DM content, lactic acid, and ammonia-N than that of CCP and SPP silages, likely due to the DRP material having high DM and WSC contents. The high WSC content indicates that it could provide more ensiling substrate, such as sugar content, to produce lactic acid and rapidly decrease pH in the silo, this low pH could inhibit the undesirable fermentation and conserve more nutrient substrate [27]. When compared to SPP and DRP silages, the CCP silages were the highest total alcohol concentration, this could be attributed to the abundance of yeast in the ensiling material ($10^9$ cfu/g FM), which utilized sugar as a substrate to produce alcohol during the fermentation process. Alcohol is one of the four main volatile organic compounds detected in silage [32]. The SPP silages were the lowest pH and highest acetic acid concentration compared to CCP and DRP silages, which could be attributed to the high number of aerobic bacteria present in the SPP material ($10^9$ cfu/g FM). Aerobic bacteria, known as acetic acid bacteria are capable of growing at low pH and cultivate acetic acid by metabolizing ethanol [33]. Our finding is consistent with Yang et al. [5] who reported that the factors involved in assessing fermentation quality include the chemical composition and the physiological properties of epiphytic bacteria of fruit residue material.

The important silage fermentation product is lactic acid since it can rapidly reduce pH, inhibit the growth of harmful microorganisms, and preserve forage nutrients [11]. Silages treated with TH14+AC and TH14+AC+molasses were lower pH and higher lactic acid content when compared with control treatments, this might be due to the synergetic effect between additives. Singh et al. [34] reported the efficacy of exogenous fibrolytic enzyme and LAB inoculant was higher when used in combinations. Similar to our findings, Kaewpila et al. [17] and Si et al. [35] found that the lactic acid content increased in cassava pulp silage and mixed of alfalfa and *Leymus chinensis* silage when treated with the combination of LAB and fibrolytic enzyme treatment.

The result found that acetic acid production increased when the additive was AC and Molasses+AC. The production of acetic acid could potentially improve the aerobic stability of silage [27]. While Guan et al. [36] reported that heterofermentative LAB produce acetic acid and their occupation of the microbial niche during terminal fermentation can enhance the aerobic stability of corn silage. For high-quality silage, Kaewpila et al. [17] suggest that the acetic acid concentration should not exceed 10–20%. For this reason, the final acetic acid contents of silages may have been related to the presence of epiphytic LAB populations in ensiling materials.

Generally, propionic acid and butyric acid is usually low or undetectable in well-fermented silages, high concentrations of propionic acid (>0.3–0.5%) are more commonly found in clostridial fermentations, and the presence of butyric acid indicates metabolic activity from clostridial organisms, which leads to large DM losses and poor recovery of energy [6]. In this study, propionic acid was highest in the control compared to other treatments, while butyric acid and ammonia-N were not significantly different among treatments. Possibly, the presence of propionic acid in control silage may be due to propionic acid bacteria converting glucose and lactic acid to propionic and acetic acids. However, the concentration of these acids and ammonia-N of tropical fruit peel silages were all within the acceptable ranges as suggested by [37]. Consequently, the use of a silage additive such as LAB inoculant, cellulase, and their combination could improve fermentation quality and inhibit the growth of harmful microorganisms.

In our study, it was observed that silage treated with AC alone or in combination with TH14 and molasses showed the highest total alcohol content compared to the other treatments. This could be attributed to the fact that AC aided in breaking down the fiber into sugar, which served as a substrate for yeast to produce alcohol. The high alcohol (ethanol) contents in the silages result from the metabolism of yeasts and heterofermentative LAB that convert the available WSC into ethanol and $CO_2$ [27,38]. Kung et al. [6] stated that the high concentrations of ethanol in silages (>3–4% on DM) are often associated with high numbers of yeasts, and such silages usually spoil readily when exposed to air, and high amounts of ethanol are also associated with high losses of DM.

### 4.3. Chemical Composition and Microbial Populations of Tropical Fruit Peel after Ensiling

In the present study, the SPP and DRP silages were higher CP content and lower NDF, ADF, and ADL contents compared to CCP silages (Table 3), this could be due to the chemical composition property of ensiling materials, which has a higher CP and lower fiber contents than CCP, indicating that SPP and DRP are appropriate for use as an alternative feed resource for ruminants. Kaewpila et al. [14] stated that the chemical compositions of crop silage were specifically affected by the silage additives. In this study, TH14+AC, Molasses+TH14, Molasses+AC, and Molasses+TH14+AC treatment increased the CP content more than the control and other treatments. This result may be clarified by the ability of the additives to rapidly decrease the pH value of silage, inhibit the activity of harmful microorganisms reduce the CP degradation and ultimately reduce the loss of nutrients [9]. The silages treated with molasses+AC, and molasses+TH14+AC not only increased CP content but also decreased NDF and ADF content. This finding is consistent with Kaewpila et al. [14] who mentioned the reduction of NDF and ADF contents with

AC is associated with enzymatic saccharification, releasing fermentable sugars for lactic acid fermentation.

Our study revealed that all tropical fruits peel silages ensiled for 30 days still had abundant LAB and aerobic bacteria, while coliform bacteria, yeast, and mold were below the detectable level (Table 4). The CCP silages showed higher LAB and aerobic bacteria populations than SPP and DRP silages ($p < 0.01$). However, silage treated with additives showed lower aerobic bacteria numbers than control treatments. These findings are attributed to the additive having the potential to improve the ensiling process [39].

### 4.4. In Vitro Rumen Fermentation

The in vitro fermentation is important for estimating energy partition potential of ruminant feedstuffs [17,40]. The physical properties of ensiling materials play a significant role in determining the digestibility efficiency of silages. In the present in vitro experiment, after 24 and 48 h of incubation, the SPP and DRP silages were significant higher IVDMD, IVOMD, and total VFAs content, while pH and ammonia-N were lower than CCP silages. (Tables 5 and 6). For this reason, it could be attributable to the CCP material having high lignin content, which is difficult to digest. These results agree with those of Hartati et al. [41], who reported that if the lignin content in the feed is high, the digestibility coefficient of the feed is low. After 24 and 48 h of incubation, silage treated with molasses was highest IVDMD and IVOMD compared to other treatments. Meanwhile, total VFAs contents was higher when silage treated with molasses and AC after incubation at 24 and 48 h, respectively. In the agreement of these results, Yildiz et al. [42] reported that silage prepared from *Brassica rapa* at the end of the flowering stage treated with molasses leads to increased IVDMD and IVOMD. Similar to our results, Dong et al. [43] and Gül [15] also reported increased IVDMD and IVOMD due to the supplementation of molasses. On the other hand, Xie et al. [9] and Wang et al. [12] reported that the IVDMD and total VFAs of alfalfa and paper mulberry silage treated with molasses did not differ with control silage. The addition of molasses enhanced digestibility may be explained by the higher residual contents of WSC, which can be utilized by rumen microorganisms for degradation during in vitro incubation [43].

### 5. Conclusions

The CCP, SPP, and DRP are abundant during the summer season, making them a valuable by-product that can be effectively preserved as silage. This presents an intriguing option for smallholder farms to utilize as animal feed during the shortage season. The present investigation showed the combination of molasses, TH14, and AC could improve the chemical compositions of tropical fruit peel silages, especially CP contents. The in vitro rumen digestibility of SPP and DRP silages was greater by 60% compared to CCP silage. The addition of molasses is the most effective on improved digestibility. However, the preservation of tropical fruit peel as silage is currently constrained by the substantial production of alcohol. Thus, further studies should investigate the utilization of a silage additive capable of reducing alcohol content. Additionally, conducting in vivo experiments is necessary to gather more data.

**Author Contributions:** Conceptualization, W.K., P.S. and C.K.; formal analysis, W.K., C.K., P.P. (Premsak Puangploy), K.K., S.T.-u. and P.S.; investigation, W.K. and C.K.; resources, W.K., P.G., A.C. and C.K.; writing—original draft preparation, W.K. and C.K.; writing—review and editing, W.K., P.P. (Premsak Puangploy), K.K., P.K., P.G., P.P. (Paiwan Panyakaew), T.K., R.S., S.T.-u., P.S., A.C. and C.K.; supervision, W.K. and C.K.; funding acquisition, C.K. All authors have read and agreed to the published version of the manuscript.

**Funding:** This research project is supported by the Science Research and Innovation Fund (Contract No. FF66-P1-022). The APC was co-funded by Rajamangala University of Technology Isan and Sakon Nakhon Rajabhat University.

**Institutional Review Board Statement:** The animal experimental protocols performed in this work were approved by the Institutional Animal Care and Use Committee of Rajamangala University of Technology Isan, based on the Ethics of Animal Experimentation of the National Research Council, Thailand (Record No. 35/2565).

**Informed Consent Statement:** Not applicable.

**Data Availability Statement:** Not applicable.

**Acknowledgments:** The authors thank Rajamangala University of Technology Isan, Khon Kaen University, and Sakon Nakhon Rajabhat University for the infrastructure and laboratory facilities. W.K. and C.K. would like to thank Yimin Cai from Japan International Research Center for Agricultural Sciences (JIRCAS) for expert technical support.

**Conflicts of Interest:** The authors declare we have no conflicts of interest. The funders had no role in the design of the study; in the collection, analyses, or interpretation of data; in the writing of the manuscript; or in the decision to publish the results.

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
