# Peer review of "In Vitro Rumen Fermentation of Coconut, Sugar Palm, and Durian Peel Silages, Prepared with Selected Additives"

_fermentation, doi:10.3390/fermentation9060567_

Round 1
Reviewer 1 Report
Dear editor,
This is an exceptionally well-done research article. The authors did a good job. I recommend the publication in its present form.
Cordially,
Author Response
Dear Reviewer 1,
The authors really appreciate thanks for your spending time in evaluating our manuscript for publication in Fermentation.
Thank you for your consideration.
Yours sincerely,
Dr. Chatchai Kaewpila

Reviewer 2 Report
The goal of the study is defined very clearly, but the practical benefits are not mentioned in the manuscript. The Authors used modern methodology and the results are presently well. The chapter "Conclusions" is recommended to improve.
Other comments are listed in attachment.

There are only minor errors and a few sentences that need to be corrected.
Author Response
Dear Reviewer 2,
The authors really appreciate thanks for your spending time in evaluating our manuscript for publication in Fermentation. Your comments have been invaluable in improving the quality of our work. We have carefully considered each of your suggestions and have made the necessary revisions to address the concerns raised. In this response, we provide a point-by-point explanation of the changes made and the clarifications we have incorporated to enhance the manuscript, please see in the attached file.
Thank you for your consideration.
Yours sincerely,
Dr. Chatchai Kaewpila

Reviewer 3 Report
The aim of the work was to determine the effects of selected ensiling additivies: LAB, cellulase enzyme, molasses, and their combinations on the course of fermentation and digestibility in vitro of silage from tropical fruit peels.
Could be a better title: In vitro rumen fermentation of coconut, sugar palm and durian peel silages, prepared with selected additives.
The aim of the work given in the abstract and in the introduction is slightly different and should be clarified.
The first paragraph of the introduction does not refer to any item of literature, but contains information that should be supported by the literature cited.
The methodological scope of the work, the content of subsequent chapters do not raise any major comments or objections.
A few comments, mostly editorial rather:
line 112: is [5], [20], should [5, 20];
line 132: is CO2, should CO2;
line 139: is 39◦C , should 39oC, and in other places also where it is ◦ instead of o;
line 155, 168, 189, and so on: there is no need to repeat the titles of tables in the text, they can be referred to by giving the table number in brackets in the next sentence.
The obtained values concerning the silages characteristic, taking into account the low content of lactic acid and the high content of NDF, ADF or alcohol, do not indicate their good quality and are differ from the requirements for "ideal silage".
The first sentence of the conclusions (line 371): „This study investigated that CCP, SPP, and DRP silage are a lactic acid fermentation”, is rather unnecesary, after all, in the ensiling process it is the proper and obvious direction of fermentation. It's better to switch the order of the sentences, first from line 373, then from 372.
The work requires corrections, which should be made before possible publication.
Author Response
Dear Reviewer 3,
The authors really appreciate thanks for your spending time in evaluating our manuscript for publication in Fermentation. Your comments have been invaluable in improving the quality of our work. We have carefully considered each of your suggestions and have made the necessary revisions to address the concerns raised. In this response, we provide a point-by-point explanation of the changes made, please see in the attached file.
Thank you for your consideration.
Yours sincerely,
Dr. Chatchai Kaewpila

Reviewer 4 Report
The research topic is interesting and might be helpful mainly to small producers. However, improving English writing is essential, as being more specific in some points related to the methodology. Also, to enhance results and discussion, complete some references that lack information.

Author Response
Dear Reviewer 4,
The authors really appreciate thanks for your spending time in evaluating our manuscript for publication in Fermentation. Your comments have been invaluable in improving the quality of our work. We have carefully considered each of your suggestions and have made the necessary revisions to address the concerns raised. In this response, we provide a point-by-point explanation of the changes made, please see in the attached file.
Thank you for your consideration.
Yours sincerely,
Dr. Chatchai Kaewpila

Round 2
Reviewer 4 Report
Significant improvements have been made to the English writing and all manuscript sections.